# Anticholinesterase Activity of Methanolic Extract of *Amorpha fruticosa* Flowers and Isolation of Rotenoids and Putrescine and Spermidine Derivatives

**DOI:** 10.3390/plants13091181

**Published:** 2024-04-24

**Authors:** Dagmar Jankovská, Nikol Jurčová, Renata Kubínová, Jiří Václavík, Emil Švajdlenka, Anna Mascellani, Petr Maršík, Kateřina Bouzková, Milan Malaník

**Affiliations:** 1Department of Natural Drugs, Faculty of Pharmacy, Masaryk University, Palackého třída 1946/1, 61200 Brno, Czech Republic; jurcova.nikol@seznam.cz (N.J.); kubinovar@pharm.muni.cz (R.K.); vaclavikj@pharm.muni.cz (J.V.); svajdlenkae@pharm.muni.cz (E.Š.); 2Department of Food Science, Faculty of Agrobiology, Food and Natural Resources, Czech University of Life Sciences Prague, Kamýcká 129, 16500 Prague, Czech Republic; mascellani@af.czu.cz (A.M.); marsik@af.czu.cz (P.M.); 3Central European Institute of Technology (CEITEC), Masaryk University, Kamenice 5/C04, 62500 Brno, Czech Republic; kamala@mail.muni.cz

**Keywords:** AChE, *Amorpha fruticosa*, BuChE, molecular docking, putrescine, spermidine

## Abstract

Five putrescine and spermidine derivatives (**1**–**5**) together with five rotenoids (**6**–**10**) were isolated from a methanolic extract of the flowers of *A. fruticosa* that displayed promising inhibition of 76.0 ± 1.9% for AChE and 90.0 ± 4.0% for BuChE at a concentration of 1 mg/mL. Although the anticholinesterase activities of the isolated compounds did not reach that of galantamine, molecular docking revealed that all-*trans*-tri-*p*-coumaroylspermidine and *trans*-*trans*-*cis*-tri-*p*-coumaroylspermidine showed binding poses mimicking the known inhibitor galantamine and thus could serve as model molecules in future searches for new AChE and BuChE inhibitors.

## 1. Introduction

*Amorpha fruticosa* L. (Fabaceae) is a shrub known by common names such as false indigo bush or desert false indigo bush. It is native to North America and is traditionally used as a blue dye. In Europe, *A. fruticosa* is known as an ornamental plant that has an extensive root system, which stabilizes the soil and protects against erosion. On the other hand, it is registered among the invasive species [1]. The scented flowers are purplish blue with orange anthers, and they bloom in May and June. The genus name *Amorpha* probably refers to the atypical structure of the flower, which has only a single petal, whereas the four other petals normally present in Leguminosae plants are completely missing [2].

The phytochemical profile of *Amorpha* species includes typical substances of the family Fabaceae, with isoflavonoids and their derivatives called rotenoids as the main group of secondary metabolites. However, unusual prenylated stilbenoids known as amorfrutins, which are present especially in the leaves and fruits, are also interesting [2]. The flower oil contains 82% sesquiterpenes, with α-eudesmol, β-eudesmol, δ-cadinene, and (*E*)-nerolidol as the dominant compounds. The types of sesquiterpenes and their relative proportions depend on the origin of the plant material [3].

Previous studies have reported the anti-tumor [4], antibacterial [5], antioxidant [6], and anti-inflammatory activities [7,8] of extracts of various parts of *A. fruticosa*. Recently, rotenoids isolated from the seeds of *A. fruticosa* displayed significant inhibitory activity against tyrosinase [9]. However, the most promising pharmacological effect of *A. fruticosa* seems to be its antidiabetic effect. Amorfrutins isolated from the fruits have been described as potent antidiabetic agents as they bind to and activate PPARγ, which results in selective gene expression and physiological profiles markedly different from those resulting from activation by current synthetic PPARγ drugs [10]. Unfortunately, a phytochemical study dealing with the biological activities and chemical constituents of the flowers is still lacking.

Phenolamides, sometimes referred as hydroxycinnamic acid amides (HCAAs) or phenylamides, have been reported as specific metabolites detected in the anthers and pollen grains. HCAAs affect plant development, adaptation, sexual differentiation, fertilization, and senescence [11,12]. Furthermore, they play an important role in the microbial protection of pollen grains [12]. Additionally, it has already been proven that phenolamides show neuroprotective properties based on their antioxidant and anti-inflammatory activities [13,14], and could therefore be beneficial in the therapy of Alzheimer’s disease (AD). AD is a progressive neurodegenerative disease, and one of the most important approaches in the multifunction therapy of AD is the inhibition of acetylcholinesterase (AChE) and butyrylcholinesterase (BuChE). The elevation of brain acetylcholine levels may reduce symptoms and help with certain behavioral problems [15]. AChE is dominant in the healthy brain, whereas BuChE plays a rather secondary role in the regulation of acetylcholine concentrations in the brain. However, BuChE activity is significantly increased in AD patients, while AChE activity remains unchanged or decreases. Both enzymes are therefore important therapeutic targets in AD patients for improving the cholinergic deficit and thus the cognitive function [16]. Therefore, this study is focused on the isolation and identification of compounds responsible for the anticholinesterase activity of the extract of *A. fruticosa* flowers.

## 2. Results

### 2.1. Isolation and Identification of Chemical Compounds and Quantification of Compound ***5***

Separation of the ethyl acetate fraction led to the isolation of five compounds (**1**–**5**). Their structures were identified by extensive NMR analysis using 1D and 2D experiments such as HSQC, HMBC, COSY, or NOESY, as well as by HR-MS analysis. Finally, spectroscopic data were compared with the literature, and the isolated compounds were identified as follows: *N*^1^-(*E*)-*N*^6^-(*Z*)-di-*p*-coumaroylputrescine (syn. mongolicine A) (**1**) [17], *N*^1^,*N*^6^-(*E*)-di-*p*-coumaroylputrescine (**2**) [18], *N*^1^-(*E*)-*N*^5^,*N*^10^-(*Z*)-tri-*p*-coumaroylspermidine (syn. safflospermidine B) (**3**) [19], *N*^1^,*N*^5^-(*Z*)-*N*^10^-(*E*)-tri-*p*-coumaroylspermidine (**4**) [20], and *N*^1^,*N*^5^,*N*^10^-(*E*)-tri-*p*-coumaroylspermidine (**5**) [21]. The presence of putrescine and spermidine conjugates in *A. fruticosa* is described here for the first time. Additionally, five rotenoids (**6**–**10**) were isolated from the chloroform fraction and their structures were elucidated by the same procedure as described above. Finally, *cis*-12a-hydroxymunduserone (**6**) [22], 6-deoxyclitoriacetal (**7**) [23], amorphispironone (**8**) [24], tephrosin (**9**) [24], and 12α-hydroxy-α-toxicarol (**10**) [25] were identified and subjected to biological assays.

Furthermore, as compound **5** is dominantly present in the crude extract and the ethyl acetate fraction, it was quantified by HPLC-DAD using the calibration curve method. Based on our calculation, a total amount of 192.9 mg of compound **5** was present in the ethyl acetate fraction.

### 2.2. Inhibition of Acetylcholinesterase (AChE) and Butyrylcholinesterase (BuChE)

The methanolic extract of the flowers of *A. fruticosa* was tested for the ability to inhibit AChE and BuChE. The process involved the extraction of 1 g of dried flowers with methanol for 24 h at room temperature, after which the extract was filtered and air-dried. The test sample was prepared by dissolving 10 mg of the extract in 1 mL of methanol. The AChE inhibition potential was quantified according to a modified Ellman’s colorimetric method, as described in Section 4.5. The methanolic extract of flowers of *A. fruticosa* exerted significant inhibitory activities of 76.0 ± 1.9% for AChE and 90.0 ± 4.0% for BuChE. Based on these results, the methanolic extract of flowers of *A. fruticosa* was subjected to isolation of the compounds responsible for the AChE and BuChE inhibitory activities. The isolated phenolamides and rotenoids were dissolved in MeOH and tested at a concentration of 100 µM. The results are summarized in Table 1.

Although the methanolic extract showed significant inhibitions of 76.0 ± 1.9% for AChE and 90.0 ± 4.0% for BuChE, the activities of the crude extract did not correspond with those of the isolated phenolamides and rotenoids. Compound **5** exhibited the highest inhibition of AChE activity, achieving 47.9% at the tested concentration of 100 µM, with an IC_50_ value of 102.0 ± 8.1 µM, the lowest IC_50_ value among all of the compounds tested. Unfortunately, the values were lower in comparison with the standard galantamine (1.1 ± 0.1 µM for AChE inhibition and 73.6 ± 0.6 µM for BuChE).

### 2.3. Molecular Docking

To support the results of bioactivity, all possible isomers of tri-*p*-coumaroylspermidine and di-*p*-coumaroylputrescine were subjected to molecular docking to confirm that their spatial orientation plays a crucial role in the inhibition of AChE and BuChE. In fact, eight tri-*p*-coumaroylspermidine conformers were modeled with the target protein structure together with four di-*p*-coumaroylputrescine conformers. The binding affinities of all possible isomers are summarized in Table 2. For comparison, galantamine and acetylcholine were also docked into the active pocket to calculate the binding affinities and poses, to be compared with the test compounds, and to compare the docked poses of all potential inhibitors in bonding to the catalytic triad of Ser200, Glu327, and His440 residues [26], as well as Glu199, where galantamine binds and is in proximity to the acetylcholine binding pose in the case of *Tetronarce californica* AChE (TcAChE), and Ser198, Glu325, and His438 in case of human BuChE (hBuChE) [27].

All possible isomers of tri-*p*-coumaroylspermidine and di-*p*-coumaroylputrescine had binding affinities to TcAChE ranging from −9.6 to −11.4 Kcal/mol, while acetylcholine showed a binding energy of −5.1 Kcal/mol. Acetylcholine is a much smaller molecule and thus has a much smaller surface interacting with the target protein structure, resulting in a lower binding energy. The binding energies of the test compounds are close to one another and about twice that of acetylcholine. The cavity leading to the enzyme’s active center is very large, and all compounds can bind into it. All-*trans*-tri-*p*-coumaroylspermidine (**5**) (Figure 1A) and *trans*-*trans*-*cis*-tri-*p*-coumaroylspermidine (Figure 1B) show binding poses that mimic the known inhibitor galantamine (Figure 1C). Other isomers also fit in the pocket and probably allow the small molecules traveling around the enzyme to remain active.

Similarly, all possible isomers of tri-*p*-coumaroylspermidine and di-*p*-coumaroylputrescine had binding affinities to hBuChE that ranged from −8.6 to −10.4 Kcal/mol, while acetylcholine showed a binding energy of −4.5 Kcal/mol. The binding energies to hBuChE were weaker than in the case of TcAChE. None of the test compounds showed additional polar contact with other residues in proximity to the active center, unlike the catalytic triad, in comparison to TcAChE. In the cases of tri-*p*-coumaroylspermidine and di-*p*-coumaroylputrescine derivatives, the binding poses differed greatly, but the all-*trans* isomer (**5**) (Figure 2A) still mimicked the galantamine pose. *Trans*-*trans*-*cis*-tri-*p*-coumaroylspermidine (Figure 2B) showed a less favorable pose.

## 3. Discussion

The presence of putrescine and spermidine conjugates in *A. fruticosa* has been described here for the first time. There is no evidence of the isolation of phenolamides from either the genus *Amorpha* or the family Fabaceae. On the other hand, these compounds have been reported in various plants across the plant kingdom, especially in bee pollen, e.g., of *Quercus mongolica* Fisch. ex Ledeb., Fagaceae [17], or of *Carthamus tinctorius* L. [20] or *Helianthus annuus* L., both Asteraceae [28].

Phenolamines are frequently substituted with cinnamic acid residues to form a group of HCAAs that play a crucial role in the development of flowers, sexual differentiation, fertilization, and senescence [11,12]. HCAAs i.a. protect pollen grains against UV irradiation and oxidative stress [11]. Very recently, Vukašinović et al. [29] quantified polyamines (putrescine, spermidine, and spermine) in honey bee products such as bee pollen, honey, royal jelly, and bee bread. Based on the palynological analysis, *A. fruticosa* was one of the minor but still important sources of bee products among many plants from various families. These results indicate that HCAAs are commonly synthesized by plants and concentrated in the flowers to protect pollen grains, and are therefore rather non-specific than could be used as chemotaxonomic markers.

Open-chain spermidine and putrescine alkaloids with phenolic acid residues exhibit a wide range of pharmacological activities, including antioxidant, anti-inflammatory, and neuroprotective properties [13,14]. A study of dicaffeoylspermidine derivatives using transgenic fruit flies (*Drosophila melanogaster*) indicated a partial recovery of short-term memory [30]. Di-*p*-coumaroylputrescine inhibited the nuclear factor NF-κB, which can stimulate the transcription of numerous genes, such as pro-inflammatory genes [31]. Di-*p*-coumaroylputrescine has also shown promising hydroxyl radical scavenging activity (IC_50_ = 164.8 ± 5.8 µM), comparable to that of α-tocopherol (IC_50_ = 155.7 ± 4.3 µM), used as a positive control. Unfortunately, di-*p*-coumaroylputrescine was inactive in DPPH and superoxide radical scavenging assays [18]. To the best of our knowledge, there is no information about the inhibition of AChE and BuChE by di- and tri-substituted spermidine and putrescine derivatives with a *p*-coumaroyl moiety. However, some macrocyclic and open-chain spermine and spermidine alkaloids have exhibited great potential for the treatment of brain diseases by inhibiting the activities of AChE and BuChE [14]. *p*-Coumaric acid alone has been shown to be a potent AChE inhibitor with an IC_50_ value of 18.7 µg/mL [15], and it also showed the highest anti-BuChE activity among the phenolic acids obtained from malt [32]. In our study, *N*^1^,*N*^5^,*N*^10^-(*E*)-tri-*p*-coumaroylspermidine (**5**) showed the highest activity of all of the compounds tested. Despite this, its effectiveness in inhibiting anticholinesterase was significantly lower compared to the positive control galantamine. Rotenoids (**6**–**10**) also showed low activities at a concentration of 100 µM, which is in good agreement with the results recently published for a novel oxidative ring-opening rotenoid that was inactive against both AChE and BuChE [33]. However, compounds **8** and **9** seem to be somehow specific to BuChE as these were completely inactive against AChE. This could be attributed to the additional pyrano ring of isoprenoid origin and possibly more favorable orientation in the space of compounds **8** and **9** in comparison to compounds **6** and **7**, lacking the additional pyrano ring. Unfortunately, the series of rotenoids isolated from *A. fruticosa* in this study was too small to reveal either specificity or a structure–activity relationship. Although spermidine and putrescine derivatives were the dominant compounds in the crude extract based on HPLC-DAD (see Appendix A), the anticholinesterase activities of the isolated compounds were poor and did not match the percentage of inhibition of the crude extract. This indicates that some content compounds probably have synergistic effects, and the crude extract therefore exerted much more pronounced AChE and BuChE inhibitory activities.

Subsequently, all possible isomers of tri-*p*-coumaroylspermidine and di-*p*-coumaroylputrescine were subjected to molecular docking to determine whether the *cis*/*trans* orientation of the *p*-coumaroyl moieties had as large of an impact on bioactivity as observed in the anti-AChE and anti-BuChE assays. Unusually, all of the modeled conformers had similar binding energies, ranging from −9.6 to −11.4 Kcal/mol in the case of TcAChE, and −8.6 to −10.4 Kcal/mol in the case of hBuChE. All of the compounds were somehow bound in the cavity of the enzyme. However, only all-*trans*-tri-*p*-coumaroylspermidine (**5**) and *trans*-*trans*-*cis*-tri-*p*-coumaroylspermidine interacted with the catalytic triad residues Ser200, Glu327, and His440 (TcAChE), and showed binding poses similar to that of galantamine. This can be attributed to the specific spatial orientation of all-*trans*-tri-*p*-coumaroylspermidine (**5**) and *trans*-*trans*-*cis*-tri-*p*-coumaroylspermidine, which is necessary for interaction with the active site of the enzyme. Tri-*p*-coumaroylspermidine and di-*p*-coumaroylputrescine derivatives partially violate Lipinski’s rule of five, mainly by the number of rotatable bonds and the low molecular weight. More rotatable bonds and thus a more flexible molecule allow more flexibility to adapt to the active site of the enzyme. This means that flexible molecules are much more sensitive to the conformational changes of the binding site, and a small conformational change in the binding site can therefore greatly affect the ligand pose. This phenomenon could be clearly observed by comparison of the results between TcAChE and hBuChE, where the active sites were similar and rigid galantamine as well as acetylcholine were docked in very similar poses, whereas the flexible test molecules differed significantly. This feature makes docking studies and interpreting docking results challenging. For better precision, the 2V97 dataset collected after annealing to room temperature was used for this docking study. A higher temperature during data collection meant a conformation closer to the enzyme in the native state of the organism. In contrast, rigid structures like galantamine are less sensitive to conformational changes of the binding pocket and are therefore more potent inhibitors of AChE and BuChE.

## 4. Materials and Methods

### 4.1. General Experimental Procedures

UV spectra were obtained using an Agilent 1100 chromatographic system with a DAD (Agilent Technologies, Santa Clara, CA, USA). HRMS data were recorded using a UPLC-HRAM-MS system consisting of q-TOF mass spectrometer Impact II (Bruker Daltonics, Bremen, Germany) coupled with a UPLC Ultimate 3000 chromatographic system (Thermo Fisher Scientific, Waltham, MA, USA), in both the positive and negative modes. We obtained 1D and 2D NMR spectra on a JEOL ECZR 400 MHz NMR spectrometer (Jeol, Tokyo, Japan) or Bruker Avance III spectrometer equipped with a broad band fluorine observation SmartProbe TM (Bruker, Billerica, MA, USA) using TMS as the internal standard. The structure of compound (**5**) was additionally verified by ^1^H NMR and DQF-COSY NMR spectra recorded on a Bruker 600 MHz Avance Neo spectrometer equipped with a room-temperature inverse quadruple resonance probe, and evaluated using TopSpin software (Bruker, Billerica, MA, USA). The signal of TMS or the residual solvent signals of CD_3_OD, CDCl_3_, or DMSO-*d*_6_ (Sigma-Aldrich, Saint Louis, MO, USA) were used for reference. Gradient-grade MeCN for HPLC was purchased from Sigma-Aldrich, USA, and other analytical-grade solvents were sourced from Lach-Ner, Czech Republic.

AChE from electric eel, BuChE from equine serum, and all reagents for enzyme inhibitory activities were purchased from Sigma-Aldrich, USA. The inhibition of AChE and BuChE was performed on a Microplate reader (Synergy HT Bio-Tek, Winooski, VT, USA).

### 4.2. Plant Material

The whole racemes of *A. fruticosa* used in this study were collected in June 2019 in a public park in Brno and identified by Dr. Jankovská. A voucher specimen (ID: AF-2019) has been deposited in the Herbarium of the Department of Natural Drugs, Masaryk University, Brno.

### 4.3. Extraction and Isolation

The air-dried flowers of *A. fruticosa* (258.8 g) were crushed before being extracted three times with 90% methanol to yield 87.5 g of crude extract. The crude extract was then successively partitioned with *n*-hexane, chloroform, and ethyl acetate.

The ethyl acetate fraction (5.576 g) was subjected to silica gel column chromatography (75 cm × 4 cm) with an isocratic elution using a mobile phase consisting of chloroform:methanol 7:3 (*v*/*v*) to afford 26 subfractions (AF_K_E_1–26). Subfractions AF_K_E_8 (68.6 mg) and AF_K_E_9 (150.1 mg) were combined and purified by semi-preparative HPLC (Dionex Ultimate 3000 system equipped with a UV detector, Dionex, Sunnyvale, CA, USA) with an Ascentis RP-Amide column (25 cm × 10 mm, 5 µm, Supelco, Bellefonte, PA, USA). The mobile phase consisted of 25% of MeCN (A) and 75% of H_2_O containing 0.2% HCOOH (B), with the gradient over 30 min reaching a composition of 52% A and 48% B. The flow rate was 5 mL/min, and UV detection was applied at 290 and 315 nm. This procedure led to the isolation of five compounds **1** (1.1 mg, *t*_R_ 11.26 min), **2** (6.3 mg, *t*_R_ 12.26 min), **3** (2.3 mg, *t*_R_ 13.80 min), **4** (3.2 mg, *t*_R_ 14.31 min), and **5** (34.2 mg, *t*_R_ 14.99 min).

The chloroform fraction (2.191 g) was separated using silica gel column chromatography with an isocratic elution using a mobile phase consisting of chloroform:methanol 94:6 (*v*/*v*) to obtain 37 subfractions (AF-K_CH_1–37). Selected subfractions were purified by semi-preparative HPLC as described above, but with different ratios of MeCN and H_2_O containing 0.2% HCOOH. UV detection was set at 280 and 350 nm. Subfraction AF-K_CH_8 (28.9 mg) was separated using gradient elution (15–78% MeCN, 30 min) to yield compounds **6** (2.5 mg, *t*_R_ 13.62 min), **7** (1.6 mg, *t*_R_ 14.93 min), **8** (1.4 mg, *t*_R_ 16.95 min), **9** (1.0 mg, *t*_R_ 18.59 min), and **10** (1.1 mg, *t*_R_ 20.03 min).

### 4.4. Quantification of Compound ***5*** by HPLC-DAD

Dominant compound **5** was quantified by HPLC-DAD using a calibration curve method, as described previously [34]. Briefly, five different concentrations (0.0625–1 mg/mL) were used to prepare a standard curve of compound **5**, constructed by a linear regression analysis. The final concentration of compound **5** in the ethyl acetate fraction was expressed as the mean of the peak areas at 280 nm of three replicates. HPLC analysis was conducted using an Agilent 1100 instrument equipped with a Diode Array Detector (DAD) (Agilent Technologies, Santa Clara, CA, USA) and an Ascentis Express RP-Amide analytical column (100 mm × 2.1 mm, particle size 2.7 μm, Supelco, Bellefonte, PA, USA). Gradient elution with the composition of the mobile phase as follows: 10–100% MeCN and 90–0% H_2_O + 0.2% HCOOH were used over 36 min. The injection volume of 1 µL was used with a flow rate of 0.3 mL/min and a column temperature of 40 °C.

### 4.5. Anticholinesterase Activity

The AChE/BuChE inhibitory activity was measured by Ellman’s spectrophotometric method, which was found in the literature [35,36] and used with slight modifications. The reaction mixture contained 120 μL of 0.1 M phosphate buffer (pH 7.0), 20 μL of 1 U/mL AChE/BuChE solution in buffer, 20 μL of the extract or a test compound, and 20 μL of 10 mM 5,5′-dithiobis-(2-nitrobenzoic acid) (DTNB). After incubation at 37 °C (15 min), 20 μL of 7.5 mM acetylthiocholine iodide (ATCI)/butyrylthiocholine iodide (BTCI) was added to initiate the reaction. The hydrolysis, which formed the yellow 5-thio-2-nitrobenzoate anion by the enzyme-catalyzed reaction of DTNB with thiocholines, was monitored at a wavelength of 405 nm. A set of mixtures prepared with an equivalent volume of water was used as a control. Another set of mixtures prepared with an equivalent volume of phosphate buffer instead of the enzyme was used as a blank. The inhibitory rates (%) were calculated according to the following formula:(1 − (A_sample_ − A_blank_)/A_control_) × 100.

### 4.6. Molecular Docking

All ligands were modeled using Marvin 23.11, 2023, ChemAxon, and the conformer with the lowest potential energy was used. Ligands were then prepared using PyRx 0.8 with the default settings [37]. PyRx 0.8 with the default settings was also used to prepare the protein structure for molecular docking. The crystal structure of *Tetronarce californica* TcAChE was downloaded from the Research Collaboratory for Structural Bioinformatics Protein Data Bank (RCSB.org). Extraneous atoms, alternate amino acid residue conformations, ligands, ions, and solvent molecules were removed. PDB ID 2v96 and 2v97 (TcAChE) and 4tpk (hBuChE) entries were used for the in silico docking experiment [26]. hBuChE 4tpk was selected since horse serum BuChE is not available in the PDB. There were no other manipulations with the protein structural conformations like energy minimization. Since data for both entries were collected at different temperatures with slightly different conformations, these differences were used to explain the in vitro experiment. PyRx 0.8 was used in conjunction with AutoDock Vina [38]. As a binding site, the entire chain A was selected in the cube-defining searching space and the exhaustiveness was set to 200. Docking experiments were performed 10 times due to the flexibility of the test compounds. PyMol [39] was used to visualize the results.

## 5. Conclusions

To the best of our knowledge, this is the first phytochemical study dealing with compounds contained in the flowers of *A. fruticosa*. An extensive separation procedure led to the isolation of five tri-*p*-coumaroylspermidine and di-*p*-coumaroylputrescine derivatives and five rotenoids. Although the anticholinesterase activities of the test compounds did not reach that of galantamine, both the results of our study and those reported earlier suggest that open-chain spermidine and putrescine alkaloids bearing the *p*-coumaroyl moiety can have protective effects on the brain in degenerative diseases or anti-AD therapy by inhibiting AChE and BuChE. Molecular docking confirmed that specific *p*-coumaroyl phenolamides interacted with the active site of the enzyme and thus could serve as model molecules for further semisynthetic modifications to improve their bioactivity. However, further studies on their pharmacological effects, bioavailability, and toxicity are required in the future.

## Figures and Tables

**Figure 1 plants-13-01181-f001:**
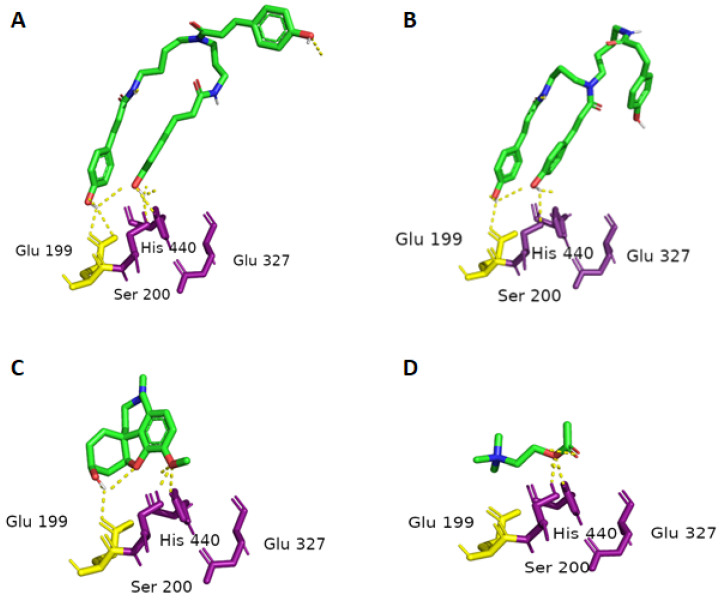
Docking poses of all-*trans*-tri-*p*-coumaroylspermidine (**A**), *trans*-*trans*-*cis*-tri-*p*-coumaroylspermidine (**B**), galantamine (**C**), and acetylcholine (**D**) in green. Dotted yellow lines visualize polar contacts with the catalytic triad residues Ser200, Glu327, and His440 in magenta. Glu199 in yellow also participates with polar contacts to galantamine and all-*trans*-tri-*p*-coumaroylspermidine stabilizing the pose with the proximity of the docked acetylcholine binding site.

**Figure 2 plants-13-01181-f002:**
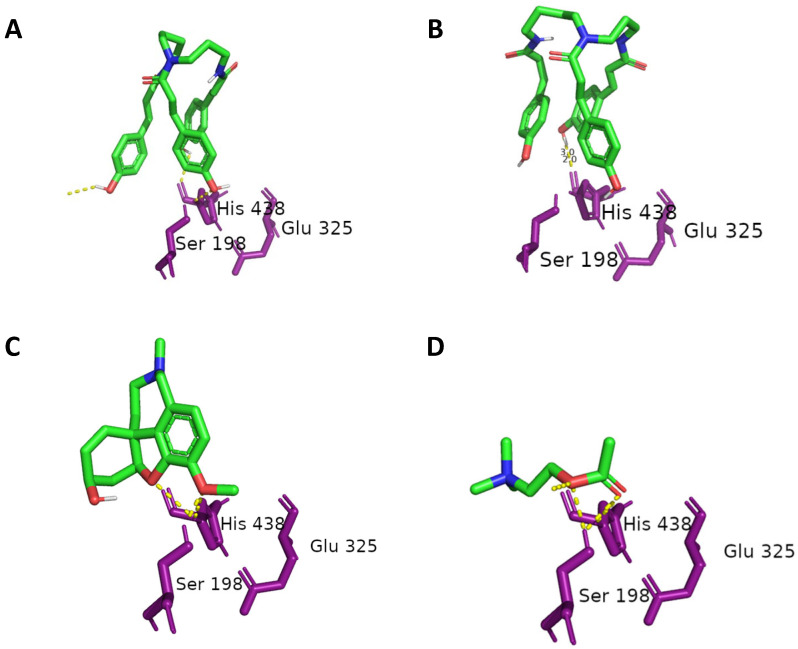
Docking poses of all-*trans*-tri-*p*-coumaroylspermidine (**A**), *trans*-*trans*-*cis*-tri-*p*-coumaroylspermidine (**B**), galantamine (**C**), and acetylcholine (**D**) in green. Dotted yellow lines visualize polar contacts, with the catalytic triad residues Ser198, Glu325, and His438 in magenta.

**Table 1 plants-13-01181-t001:** Inhibitory potency of isolated phenolamides, rotenoids, and galantamine against AChE and BuChE at a concentration of 100 µM.

Compound	% Inhibition of AChE	% Inhibition of BuChE
**1**	9.1 ± 3.2	8.1 ± 5.9
**2**	1.1 ± 0.7	4.8 ± 3.3
**3**	14.1 ± 3.4	4.5 ± 2.5
**4**	26.6 ± 2.5	19.1 ± 2.1
**5**	47.9 ± 2.1	43.8 ± 1.9
**6**	17.9 ± 5.9	23.8 ± 5.2
**7**	37.9 ± 5.0	25.6 ± 5.6
**8**	N.I.	37.6 ± 2.7
**9**	N.I.	23.6 ± 2.6
**10**	22.4 ± 4.9	22.8 ± 5.9
galantamine	82.1 ± 0.9	62.8 ± 0.3

N.I. = no inhibition. Values are expressed as the mean ± SD of three replicates.

**Table 2 plants-13-01181-t002:** Binding affinities of all possible isomers of tri-*p*-coumaroylspermidine and di-*p*-coumaroylputrescine to AChE and BuChE.

Compound	Binding Affinity to AChE [Kcal/mol]	Binding Affinity to BuChE [Kcal/mol]
**trans-trans-trans**	−10.8	−9.4
**trans-trans-cis**	−10.4	−9.6
**trans-cis-trans**	−10.8	−10.4
**trans-cis-cis**	−11.3	−10.3
**cis-trans-trans**	−11.3	−10.2
**cis-trans-cis**	−11.3	−9.5
**cis-cis-trans**	−11.4	−9.9
**cis-cis-cis**	−11.2	−10.0
**trans-trans**	−9.8	−8.4
**trans-cis**	−9.6	−9.6
**cis-trans**	−10.0	−8.7
**cis-cis**	−10.1	−8.5
Galantamine	−10.5	−8.6
Acetylcholine	−5.1	−4.5

## Data Availability

Data generated in this research are available from the authors.

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
