# Peer review of "Anticholinesterase Activity of Methanolic Extract of Amorpha fruticosa Flowers and Isolation of Rotenoids and Putrescine and Spermidine Derivatives"

_plants, 2024, doi:10.3390/plants13091181_

Round 1
Reviewer 1 Report
Comments and Suggestions for Authors
The study reports isolation and anticholinesterase screening of several hydroxycinnamylamides (mainly spermidine derivatives) from Amorpha fruticosa, a plant known to contain several classes of pharmacologicallly relevant phytochemicals. Here, the Authors used flowers (see below for comment) as source of the 10 isolated compounds that have been identified as known compounds (not from Amorpha, though) 1-10 and subsequently tested for in vitro inhibition of AChE and BChE using simple Ellmann's assay. A molecular docking of the structures to enzymes models was also performed.
The study is original as amorpha flowers phytochemistry is little studied and deserves publication in PLANTS. The methods used are technically correct and adequate. The results are interpreted reasonable and discussed insightfully.
However, there are still some additional analyses to be performed as well as some explanation to the content:
1. Please, perform an HPLC quantitation of the identified compounds in the crude extract. I know that it may be tricky for compounds isolated in 1 mg or so. Maybe you still have more material. It is essential to verify the feasibility of the approach and estimate recovery. A minimum would be to determine the compound 5 (>30 mg - enough for calibration, and use it as a standard also for other amine peaks and the rotenoids can be for example isolated from vegetative parts.). Also, the peaks in the supplementary file should be identified.
2. A question - why exactly this extraction method? Has it been optimized before, using any experimental design (RSM or something like that)? Or at least some literature evidence that 90% MeOH and maceration is a good recovery approach.
3. A comment regarding parts used - did you pick flowers separately from the inflorescence or used whole racemes ? The flowers are tine and delicate, so please, explain.
4. Please, correct the phase B description - it sounds like 75% of 0.2% formic acid ?!
5. How were the test compounds dissolved for ChE assay?
6. Please, provide manufacturers and quality for all reagents and chemicals. Also, the origin of cholinesterases is crucial for interpretation.
7. Please, draw all structures and attach as supplementary material.
8. The compounds 8 and 9, even if weak inhibitors, seem to be somehow specific to BChE, please try to discuss it.
Author Response
Dear Reviewer 1,
we appreciate your valuable comments leading to the improvement of our manuscript. We addressed all required issues and highlighted changes in the manuscript and supplementary material in yellow.
1. Please, perform an HPLC quantitation of the identified compounds in the crude extract. I know that it may be tricky for compounds isolated in 1 mg or so. Maybe you still have more material. It is essential to verify the feasibility of the approach and estimate recovery. A minimum would be to determine the compound 5 (>30 mg - enough for calibration, and use it as a standard also for other amine peaks and the rotenoids can be for example isolated from vegetative parts.). Also, the peaks in the supplementary file should be identified.
Response: Dominant compound 5 was quantified by HPLC-DAD using a calibration curve method. Based on our calculation, the total amount of 192.9 mg of compound 5 is present in the ethyl acetate fraction. We did not express the total amount of compound 5 in the crude extract as the crude extract was of oily consistency and we did not dry it but immediately stepped to the liquid-liquid extraction. However, no phenolamides have been detected in n-hexane, chloroform, or water fractions. We therefore believe that the amount determined in the ethyl acetate fraction could correspond to the amount present in the crude extract. Unfortunately, as we are not able to prove this hypothesis, we express the total amount of compound 5 in the ethyl acetate fraction.
The identification of the peaks was added to the supplementary file.
2. A question - why exactly this extraction method? Has it been optimized before, using any experimental design (RSM or something like that)? Or at least some literature evidence that 90% MeOH and maceration is a good recovery approach.
Response: The extraction method has not been optimized before use. Mixtures of alcohol–water are recommended in cases of the extraction of polar compounds (e.g. phenolic acids, (iso)flavonoid glycosides) as pure organic solvents are not able to completely extract these compounds (e.g. 10.3390/molecules21070901). We dealt with the extraction of polyphenolic substances, and therefore, we chose the extraction agents with a small percentage of water to maximize the extraction of polyphenols (e.g., (iso)flavonoid glycosides) that are typical for A. fruticosa. The isolation of polyamines was not the initial plan for the extraction as nitrogen-containing compounds have never been described in A. fruticosa before.
3. A comment regarding parts used - did you pick flowers separately from the inflorescence or used whole racemes? The flowers are tine and delicate, so please, explain.
Response: You are right that the flowers are very tiny and that is also the reason that we used the whole racemes for the extraction. The information was added to the text.
4. Please, correct the phase B description - it sounds like 75% of 0.2% formic acid ?!
Response: We apologize for the mystification. The composition of the mobile phase was clarified in the text.
5. How were the test compounds dissolved for ChE assay?
Response: Compounds were dissolved in MeOH and tested at a concentration of 100 µM. The information was added to the text.
6. Please, provide manufacturers and quality for all reagents and chemicals. Also, the origin of cholinesterases is crucial for interpretation.
Response: Requested information was added to the text. For in vitro testing, horse serum BuChE and Tetronarce californica TcAChE enzymes were used because of their availability. In the case of BuChE, the Protein Data Bank contains seven records, and all have Humans as a source organism. There were no other choices. In the case of TcAChE, 2V96 and 2V97 records were used for comparison. Both datasets were collected at different temperatures. 2V96 was a standard collection at 100K, while 2V97 was collected after annealing to room temperature. We expect the 2V97 dataset collected near room temperature to have conformation closer to the native state in living organisms since Tetronarce californica prefers 10–13°C. Because of the high flexibility of the test compounds, we observed that this difference seems to be important for molecular docking interpretation. In the case of rigid molecules like galantamine, the difference is not so evident.
7. Please, draw all structures and attach as supplementary material.
Response: The structures of isolated compounds were added to supplementary material.
8. The compounds 8 and 9, even if weak inhibitors, seem to be somehow specific to BChE, please try to discuss it.
Response: A short discussion was added to the text. However, it is very difficult to reveal any specificity of rotenoids based on a small series of compounds isolated in this study.
Reviewer 2 Report
Comments and Suggestions for Authors
- Discuss the presence of putrescine and spermidine derivatives in the genus and the family. It appears that the paper is reported these kinds of compounds in the genus for the first time.Thus, the results could be interesting also from a chemotaxonomy perspective
- Add the concentration of galantamine in Table 1. If it was used at 100 µM like the test compounds, change the legend to "Inhibitory potency of isolated phenolamides, rotenoids and galantamine against AChE and BuChE at a concentration of 100 µM."
- Italicize A.fruticosa in lines 224 , 229 and later when it was not
- Clarify if an isocratic system or gradient system was used for column chromatography. If a gradient system was used, please indicate the chloroform:methanol ratio for the first and last mobile phase in lines 233 and 244.
Author Response
Dear Reviewer 2,
we appreciate your valuable comments leading to the improvement of our manuscript. We addressed all required issues and highlighted changes in the manuscript and supplementary material in yellow.
- Discuss the presence of putrescine and spermidine derivatives in the genus and the family. It appears that the paper is reported these kinds of compounds in the genus for the first time. Thus, the results could be interesting also from a chemotaxonomy perspective.
Response: Discussion was enlarged based on your recommendation.
- Add the concentration of galantamine in Table 1. If it was used at 100 µM like the test compounds, change the legend to "Inhibitory potency of isolated phenolamides, rotenoids and galantamine against AChE and BuChE at a concentration of 100 µM."
Response: Both isolated compounds and galantamine were used at a concentration of 100 µM. The legend of Table 1 was corrected based on your suggestion.
- Italicize A. fruticosa in lines 224, 229 and later when it was not
Response: We apologize for our carelessness. The name of the plant is italicized throughout the text now.
- Clarify if an isocratic system or gradient system was used for column chromatography. If a gradient system was used, please indicate the chloroform:methanol ratio for the first and last mobile phase in lines 233 and 244.
Response: We used an isocratic system for the separation of both the ethyl acetate fraction and the chloroform fraction. This information was added to the text.
Reviewer 3 Report
Comments and Suggestions for Authors
"The manuscript "Anticholinesterase activity of methanolic extract of Amorpha fruticosa flowers and isolation of rotenoids and putrescine and spermidine derivatives" discusses the search for natural substances with anticholinesterase activity that would be useful in Alzheimer's therapy.
The topic is interesting, but it is presented quite succinctly. The authors have isolated and identified certain substances with known properties. However, it is necessary to add information about their amounts in the methanol extract once it showed the highest inhibitory activity. Thus, it will be assessed in what ratio they provide this effect. In addition, it would be useful to add the results of the study of the inhibitory activity of the remaining 18 plant extracts, for example, in the supplementary materials. The IC50 of some of the substances is mentioned in the text. Apparently there is data for everyone and let it be added. For clarity, it is good to show the structures of the isolated substances as well, although some of the isomers are visible in the Molecular Docking section. All this will contribute to a more comprehensive discussion. The text from line 85 to line 89 is more appropriate for the Materials and Methods section. There is no mention of the hexane fraction - what is in it, is it developed like the other two."
Comments on the Quality of English Language"The manuscript "Anticholinesterase activity of methanolic extract of Amorpha fruticosa flowers and isolation of rotenoids and putrescine and spermidine derivatives" discusses the search for natural substances with anticholinesterase activity that would be useful in Alzheimer's therapy.
The topic is interesting, but it is presented quite succinctly. The authors have isolated and identified certain substances with known properties. However, it is necessary to add information about their amounts in the methanol extract once it showed the highest inhibitory activity. Thus, it will be assessed in what ratio they provide this effect. In addition, it would be useful to add the results of the study of the inhibitory activity of the remaining 18 plant extracts, for example, in the supplementary materials. The IC50 of some of the substances is mentioned in the text. Apparently there is data for everyone and let it be added. For clarity, it is good to show the structures of the isolated substances as well, although some of the isomers are visible in the Molecular Docking section. All this will contribute to a more comprehensive discussion. The text from line 85 to line 89 is more appropriate for the Materials and Methods section. There is no mention of the hexane fraction - what is in it, is it developed like the other two."
Author Response
Dear Reviewer 3,
we appreciate your valuable comments leading to the improvement of our manuscript. We addressed all required issues and highlighted changes in the manuscript and supplementary material in yellow.
1. It is necessary to add information about their amounts in the methanol extract once it showed the highest inhibitory activity. Thus, it will be assessed in what ratio they provide this effect.
Response: We agree with your suggestion, Therefore, quantification of dominant compound 5 was added to the text.
2. It would be useful to add the results of the study of the inhibitory activity of the remaining 18 plant extracts, for example, in the supplementary materials.
Response: Initially, we preferred not to publish the results of bioactivity of further plant extracts as some of these are currently subjected to the isolation of content compounds and others will be subjected in the near future. We added a table summarizing the results of bioactivity of further plant extracts to supplementary material. If you insist on the attachment of this table, we will keep it as a part of supplementary material. Otherwise, we prefer not to publish the results of further plant extracts.
3. The IC50 of some of the substances is mentioned in the text. Apparently there is data for everyone and let it be added.
Response: In the case of IC50, it is common to express the IC50 of only those substances exceeding 50% of inhibition. That is the reason why we did not evaluate IC50 of further compounds.
4. It is good to show the structures of the isolated substances as well.
Response: The structures of isolated compounds were added to supplementary material.
5. The text from line 85 to line 89 is more appropriate for the Materials and Methods section.
Response: As we did not discuss the extraction and evaluation of bioactivity of further extracts, we prefer to keep the text from line 85 to line 89 as a part of the main text.
6. There is no mention of the hexane fraction - what is in it, is it developed like the other two."
Response: The crude extract and all fractions obtained by liquid-liquid extraction were analyzed by HPLC-DAD and HPLC-ELSD. The hexane fraction contained only lipophilic compounds not absorbing UV and thus was not attractive for us. For sure, it will be also beneficial to focus on the profiling of the hexane fraction to reveal if its content compounds may contribute to the AChE/BuChE inhibition.
Round 2
Reviewer 1 Report
Comments and Suggestions for Authors
The revised version has been improved according to my suggestions.
Specifically, the HPLC quantitation of selected isolates, peak annotation, methodological details, and extended discussion were added.
Thank you and I recommend accepting this paper for publication in the Plants, section Phytochemistry.
Author Response
Dear Reviewer 1,
thank you for your expert insight and evaluation of our manuscript.
Reviewer 3 Report
Comments and Suggestions for Authors The authors have made additions to the text and clarificationswhere necessary, improving the quality of the manuscript.
As for the testing of 20 plant extracts for inhibitory activity,
I understand that the authors prefer not to show this data.
Then let's remove the sentence that mentions this information
from part 2.2.
Author Response
Dear Reviewer 3,
thank you for your expert insight and evaluation of our manuscript. We also appreciate your willingness not to necessarily publish the data of the inhibitory activity of further plant extracts.